# Mental Health of Nurses Working in a Judicial Psychiatry Hospital during the COVID-19 Pandemic in Italy: An Online Survey

**DOI:** 10.3390/medsci10040061

**Published:** 2022-10-24

**Authors:** Gianluca La Rosa, Maria Grazia Maggio, Antonino Cannavò, Daniele Tripoli, Federico Di Mauro, Carmela Casella, Giuseppe Rao, Alfredo Manuli, Rocco Salvatore Calabrò

**Affiliations:** 1AOU Policlinico G. Martino, 98125 Messina, Italy; 2Department of Biomedical and Biotechnological Science, University of Catania, 95123 Catania, Italy; 3Azienda Ospedaliera Papardo, 98158 Messina, Italy; 4IRCCS Centro Neurolesi Bonino Pulejo, 98121 Messina, Italy

**Keywords:** burnout, Maslach Burnout Inventory, work stress, prisons

## Abstract

The onset of this new pandemic has highlighted the numerous critical issues at the organizational level, which involve both national healthcare and the judicial system. For this reason, nurses working in prisons may exhibit a poor quality of life, mainly related to their high level of work stress. This cross-sectional survey aimed to assess the emotional state of nurses working in the Judicial Psychiatry Hospital of Barcellona PG (Messina, Italy) during the COVID-19 pandemic. Data collection occurred twice: from 1 April to 20 May 2020 (i.e., during the Italian lockdown) and from 15 October to 31 December 2021 (during the second wave). At baseline, the 35 enrolled nurses presented medium to high levels of stress. At T1, they had a reduction in perceived personal achievement (MBI-PR *p* = 0.01), an increase in emotional exhaustion (MBI-EE *p* < 0.001), and stress (PSS *p* = 0.03), as well as anxiety (STAI Y1/Y2 *p* < 0.001). Most participants underlined the high usability of the online system (SUS: 69.50/SD 19.9). We also found increased stress, anxiety, and burnout risk in nursing staff. The study clearly demonstrates that the first year of the COVID-19 pandemic in Italy caused a worsening of mental health among nurses working in prisons. We believe that monitoring the mental state of healthcare professionals is fundamental to improving their quality of life and healthcare services.

## 1. Introduction

The beginning of 2020 was characterized by an unprecedented pandemic that affected populations worldwide. It has been shown that epidemics and outbreaks of contagious diseases have been followed by drastic individual and social, psychosocial impacts, which eventually become more pervasive than the epidemic itself [1]. During the pandemic, the population had to respect social distancing and the reduction in daily activities so as to reduce the risk of new infections. On the other hand, healthcare professionals had higher workloads and professional responsibilities. The onset of this pandemic has highlighted the numerous criticalities at the organizational level involving both the national healthcare and the judicial systems [2,3]. Inside the prisons, inmates and staff share a confined environment that could act as a reservoir for epidemics. Indeed, detention facilities are extremely susceptible to infectious diseases’ rapid and disastrous spread [4,5]. Inmates have a high prevalence of chronic mental illnesses also related to their aging. In fact, since the beginning of the pandemic, the prison population has had higher COVID-19 cases and death rates than the general population, also due to a high prevalence of previous medical conditions [6,7]. The staff working in prisons are exposed to numerous safety problems in assisting patients with drug and alcohol addiction and aggressive personalities, especially in judicial psychiatry hospitals. For this reason, prison nurses may exhibit a poor quality of life related to work stress compared to their non-prison peers [8]. Prison nurses and doctors were more exposed to emotional exhaustion and burnout, not only to direct contact with people (inmates, multidisciplinary teams, prison guards) but also to COVID-19, which has led to great uncertainty and a sense of helplessness. Healthcare personnel faced an unknown situation, aggravated by the presence of psychosocial risk factors, such as long work shifts, emergency/urgency management without knowledge of the case, and high emotional load deriving from constant contact with high-profile situations [8,9]. This set of factors has caused general uncertainty, making prisons a potentially work-related stress risk context and burnout [3,10]. Burnout is a syndrome that causes suffering and physical, mental, psychological, and/or social dysfunctions, which occur when the demands coming from the job are not adequate to the skills, resources, or needs of the worker [11]. Burnout is often found in demanding, high-contact work environments with people such as teachers and healthcare professionals [12,13,14]. According to Maslach and Jackson [15], it is characterized by (i) emotional exhaustion, i.e., the feeling of being emotionally emptied and drained from the relationship with others; (ii) depersonalization, with negative and cynical attitudes toward users; and (iii) reduced personal fulfillment, as a feeling of inadequacy and low professional self-esteem. In this regard, “helping professions”, including nurses, have an emotionally demanding relationship with clients in a state of difficulty and in need of care; this can negatively affect their well-being, especially during the pandemic.

Thus, this study aimed to evaluate the emotional state and work-related stress of nurses working in a judicial psychiatry hospital during the COVID-19 pandemic.

## 2. Materials and Methods

### 2.1. Study Design and Population

We used a cross-sectional survey design to assess the psychological response of nurses working in a judicial psychiatry hospital located in Barcellona Pozzo Gotto (Messina), Sicily, Italy, during the first year of the COVID-19 pandemic, using an anonymous online questionnaire. 

The final sample consisted of 35 nurses (Table 1). To be included in the study, nurses had to have worked for at least 1 year in the institute, and they should not have a second job.

This study was conducted in accordance with the 1964 Helsinki Declaration and subsequent amendments. All participants provided informed consent to enter the study protocol.

### 2.2. Procedures

Following the restrictive measures adopted by the Italian Government to deal with the pandemic, given that it was necessary to minimize face-to-face interactions and stay at home, we asked participants to fill out the online questionnaire. The online survey has been administered through the CAWI (Computer Assisted Web Interviewing) method: the invitation to the questionnaire has been sent through the technological means offered by smartphones (i.e., WhatsApp, Facebook) or by e-mail. They completed the questionnaires in Italian through an online survey platform (“Google Form”, Google LLC). Data collection occurred from 10 March to 20 May 2020, during the first Italian lockdown, and from 1 May to 20 September 2021, during the national emergency phase.

### 2.3. Survey Development

The questionnaire included three areas that collected closed-ended questions with evaluation on 5-point Likert scales and binary types (except for the first one that collected sociodemographic data).

The survey consisted of:(i)Sociodemographic data (sex, age, education, marital status);(ii)Tools investigating the physical and mental health of the participants:(a)The Revised Impact of Event Scale—IES [16];(b)The Depression Anxiety Stress Scales (DASS–21) [17].
(iii)The Maslach Burnout Inventory (MBI) [15];(iv)A tool on the usability of the online survey, namely the “System Usability Scale” (SUS) [18]. For more details, see Table 2.

### 2.4. Statistical Analysis

The descriptive statistics were analyzed and expressed as mean ± standard deviation or as median ± first third quartile for continuous variables, as appropriate; frequencies (%) were used for categorical variables. Clinical scale scores were expressed as a mean and standard deviation; the perception of the usability of the questionnaire was expressed in percentages. The normality of the variables was analyzed using the Kolmogorov–Smirnov test. Since most of the target variables were non-normal distributed, a non-parametric analysis was performed. Thus, the Wilcoxon signed-rank test was used to compare the group between baseline and the end of the study (intra-group analysis). Statistical analysis was performed using SPSS Statistic 18.0 (IBM SPSS Statistics, New York, NY, United States), considering a *p* < 0.05 statistically significant.

## 3. Results

Of the 48 prison nurses contacted, only 40 responded to the survey. Unfortunately, 5 subjects were lost at T1, so the final sample consisted of 35 nurses. The results showed significant differences between T0 and T1, as shown in Table 3. In particular, the results of the Wilcoxon signed-rank test underlined significant changes in the indices linked to burnout syndrome. Indeed, at T1, we found a reduction in the perceived personal achievement index (*p* = 0.01) or a lower perception of one’s competence and the desire for success in working with others. An increase in emotional exhaustion (*p* < 0.001) related to the sense of emotional drying up and exhaustion due to work was also detected. In addition, both stress level (*p* = 0.03) and the anxiety symptoms (*p* < 0.001) increased. It is worth noting that even at baseline, medium-high stress levels were found in the healthcare professionals (the rate obtained from the PSS test exceeds the cut-off level compared to the normative population). Finally, we observed a high level of satisfaction with the use of the online form for the survey. Therefore, the use of the online tool should not have affected the answers provided (69.50/SD 19.9).

## 4. Discussion

This study evaluated the mental health of nurses working in a judicial psychiatric hospital during the COVID-19 era. Various authors found that the COVID-19 pandemic exacerbated the psychological distress of healthcare workers, including stress, anxiety, depression, burnout, post-traumatic stress symptoms, and sleep disturbances [19,20,21,22,23]. In particular, Tabur et al. [23] demonstrated that several factors during the pandemic stimulated the desire to leave work, such as prolonged exposure to patients with severe conditions, irregular working hours, and workload. These aspects are very evident in prison structures as well as in justicial psychiatry hospitals [24]. The condition of nurses in prison is not well investigated. However, even in the pre-pandemic phase, it has been shown that working in prisons is usually linked to high levels of stress and burnout [25,26,27]. In fact, a study by Guardiano et al. [28] compared prison nurses and community nurses. Although the differences were not statistically significant, they found high occupational stress and a prevalence of post-traumatic stress symptoms in both groups. However, prison nurses were most affected by the increase in working load. In a recent cross-sectional survey of 589 prison workers, a high prevalence of psychological symptoms was also found [29]. According to the authors, these rates are higher than those reported in most studies of hospital healthcare workers [29].

Following the pandemic, to ensure the safety of detainees and staff, various rules that have changed the lives of staff and detainees, such as meetings with family members, communication between detainees and staff, and procedures for new assumptions, have been adopted. This situation has triggered interpersonal, managerial, and operational difficulties, with high discomfort for staff and prisoners [10,24]. According to these studies, we observed how the perceived stress during the first year of the pandemic affected psychological well-being both at work and in personal life since we found an increasing level of stress and anxiety. The nurses also presented a high risk of burnout, with an increase in emotional exhaustion or a feeling of being emotionally drained from the relationship with others; reduced personal achievements, such as feelings of inadequacy and low professional self-esteem, were also found.

However, it is noteworthy that the mental health of nurses worsened over time (from T0 to T1), although lockdown restrictions with their consequences were heavier at T0. Then, one would have expected nurses to have an improvement in psychological well-being as a consequence of the reduction in social restrictions before and during the second wave (at T1). It is possible that after one year of facing the pandemic, healthcare professionals, including nursing, may have accumulated work stress, anxiety, and other psychiatric problems with the exhaustion of coping strategies and burnout. In fact, according to Gee et al., it is also interesting to note that the nursing workforce was already experiencing a collapse before the pandemic as a result of the fact that they often experience traumatic stress, cumulative pain, and moral suffering [20]. In our sample, in fact, we found a medium-high level of stress also in the pre-pandemic phase, which greatly increased in the COVID-19 period leading to burnout. This issue is very important because burnout and secondary traumatic stress can lead to medical/caring errors and affect patient standards of care, particularly compromising compassionate care [14,22,30]. Indeed, according to Cooper and Marshall, burnout may cause individual and organizational diseases, such as serious accidents, inefficiency, and frequent absences [31]. Thus, burnout has repercussions both on the healthcare professions, causing a deterioration in the quality of life and the service offered, and on the health system, with a significant increase in health, legal and social security costs [32,33]. To this end, our study highlights that prison nursing may present multiple criticalities, which affect the personal and organizational levels, with potentially negative repercussions both on prisoners and on healthcare professionals’ relationships. Therefore, it would be useful to identify measures to better assess and improve the quality of life of healthcare personnel.

The present study had some limitations. First of all, we had to use new technologies to administer the survey for the COVID-19 restrictions, although nurses were not trained to use the tool. We are unaware whether the use of these tools could have influenced the results, although the participants did not report difficulties in using them. Future studies could investigate the use of telemedicine as an assessment tool for healthcare staff. Additionally, the study involved a small sample of nurses, so it is difficult to generalize the results to the population. However, we focused only on one of the few psychiatric hospitals present in our country, and the results are rather homogeneous. Another limitation is the lack of long-term follow-up, and it is not sure if and to what extent the results obtained would have lasted over time, also in consideration of the persistence of the COVID-19 emergency, with alternating phases of lockdown and reduction in restrictions. Although this two-point survey has followed nurses for about one year, future studies comparing the prolonged effect of exposure to this “enduring” pandemic are needed. Furthermore, we did not assess the cognitive and physical status of the nurses, as we focused exclusively on their mental status. Finally, we did not collect data on inmates and other staff present in the facility because they are outside the scope of the research. In future research, it will be necessary to extend the study to a larger sample and increase the involvement of the remaining healthcare and non-health personnel, possibly extending the data to the prisoners/psychiatric patients living in the facility. In addition, it would be better to insert indices of the mental and working state of the operators to encourage knowledge of the phenomenon and exclude other factors that may affect the results.

## 5. Conclusions

This study assessed the mental health of nurses working in a judicial psychiatry hospital during the first phase of the COVID-19 pandemic in Italy. We found increased stress, anxiety, and burnout risk in the nursing staff. We believe that innovative tools, such as online questionnaires, could be a valid solution to monitor the mental state of healthcare personnel exposed to multiple factors that cause a worsening in personal and working quality of life. Indeed, monitoring these important issues in prisons may help in better managing work-related problems with better well-being for both nurses and prisoners. Therefore, in addition to organizational and structural improvements to the prison system (e.g., more adequate shifts, less overcrowding etc.), psychological support for health workers could be a valid way to promote psychological well-being in these healthcare workers. In fact, it is necessary to evaluate and adopt strategies that improve the quality of work and the well-being of nursing staff working in prisons, given that the current malaise could have serious repercussions on healthcare professionals, especially during pandemics.

## Figures and Tables

**Table 1 medsci-10-00061-t001:** Sociodemographic characteristics of the sample.

Sociodemographic Variables	Value
Sex	
Male	19 (54.2%)
Female	16 (45.7%)
Age (Years)	41.77 ± 11.42
Social Status	
Single	20 (57.1%)
Married	15 (42.8%)

Mean ± standard deviations were used to describe continuous variables; proportions (%) were used to describe categorical variables.

**Table 2 medsci-10-00061-t002:** Clinical assessment tools.

Test/Scale	Description
DASS-21	The Depression Anxiety Stress Scales (DASS-21) is a standardized questionnaire validated in the Italian context that evaluates the emotional states of depression, anxiety, and stress. The questionnaire consists of 21 questions, with the possibility of 4 responses based on the frequency with which the subject has experienced the sensations described (0 = It has never happened to me; 1 = It has happened to me a few times; 2 = It has happened to me with a certain frequency; 3 = It almost always happened to me). An example of a question is “I felt a lot of tension and had difficulty recovering a state of calm” or “I felt very wheezy with difficulty breathing (e.g., very fast breathing, feeling of strong wheezing in absence of physical effort)”.
MBI	The Maslach Burnout Inventory (MBI) is a standardized questionnaire validated in Italian. The questionnaire is based on 22 items, each with 6 types of answers based on how the subject feels (0 = never; 1 = sometime a year; 2 = once a month or less; 3 = sometime a month; 4 = once a week; 5 = a few times a week; 6 = every day) and is designed to assess an individual’s burnout level. Based on the answers, it is possible to derive three scales related to burnout: (i) emotional exhaustion, i.e., the feeling of being emotionally emptied and drained from the relationship with others; (ii) depersonalization, with negative and cynical attitudes toward users; and (iii) reduced personal fulfillment, as a feeling of inadequacy and low professional self-esteem. An example of an item is “I feel emotionally exhausted from my job.”
The Revised Impact of Event Scale–IES	Impact of Event Scale-Revised (IES-R) is a standardized self-assessment measure that evaluates the subjective distress caused by traumatic events. The questionnaire comprises 22 items, with 4 possible answers relating to the frequency with which the subject has thought about what is stated in the sentence (1 = never; 2 = rarely; 3 = sometimes; 4 = often). Examples of items are: “I thought about the traumatic event even though I didn’t intend to” or “I had difficulty falling asleep or staying asleep, due to images or thoughts related to the traumatic event returning to me in the mind“.
SUS	The system usability scale (SUS) is a standardized scale composed of 10 items with a dichotomous (yes/no) response, which evaluates the subject’s perception of the usability of the online tool to respond to the survey. An example of items is: “The questionnaire administration method seemed complicated to fill in”.

**Table 3 medsci-10-00061-t003:** Wilcoxon signed-rank test of neuropsychological evaluation.

	T0 Median (First-Third Quartile)	T1 Median (First-Third Quartile)	*p*-Value
MBI-EE	15.5 (15.0–19.0)	26.0 (25.0–29.0)	0.00
MBI-DP	0.4 (0.0–1.0)	0.68 (0.0–1.0)	0.96
MBI-PR	36.0 (31.0–40.0)	23.0 (22.2–18.0)	0.01
PSS	12.4 (10.0–16.0)	24.0 (25.2–19.7)	0.03
STAI-Y1	34.0 (31.0–35.0)	47.0 (36.2–49.5)	0.00
STAI–Y2	30.0 (21.0–34.0)	45.60 (31.7–49.7)	0.00

Significant differences are in bold. Legend: Maslach Burnout Inventory (MBI): Emotional Exhaustion (EE; cut-off > 23), Depersonalization (DP; cut-off > 8), Personal Realization (PR; cut-off < 29); Perceived Stress Scale (PSS; cut-off < 12); State-Trait Anxiety Inventory (STAI) Y1 (state; cut-off > 36.2), and Y2 (trait; cut-off > 37.1).

## Data Availability

Data can be obtained on request to the corresponding author.

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
