# Peer review of "Mental Health of Nurses Working in a Judicial Psychiatry Hospital during the COVID-19 Pandemic in Italy: An Online Survey"

_medsci, 2022, doi:10.3390/medsci10040061_

Round 1

Reviewer 1 Report

Introduction:

The literature review and presentation is not appropriate. In the third year of the pandemic, a thorough overview of early reports, large (outcome) studies and complex data analysis is recommended. Although the authors provide some of all of these studies in the reference section, a clear research question is missing. The authors refer to the "2019 report ..." with a false reference!

Methods:

The survey development section is confusing, a clear description of all survey elements in the table is necessary and there are several misleading explanations (for example "5-point Likert scales" and then "Likert scale with three items"). Some population data are show in the table, relevant data are missing and age/sex are mentioned several times (methods, table, and results).

Results:

The data presentation, especially the abbreviations, is confusing and incomplete. According to previous published studies, for example reference 13 (Serrao C et al.), a clear survey framework with subsequent consistent presentation of all survey results is mandatory.

Discussion:

The authors should strictly note the following argument chain in their discussion (in the context of early reports, large studies and complex data analysis): what is already known and published, their own findings and how their findings contribute to existing knowledge.

Author Response

Introduction:

The literature review and presentation is not appropriate. In the third year of the pandemic, a thorough overview of early reports, large (outcome) studies, and complex data analysis is recommended. Although the authors provide some of all of these studies in the reference section, a clear research question is missing. The authors refer to the "2019 report ..." with a false reference!

Thanks for the suggestion, the error concerning the “false ref” has been removed and the literature has been updated.

Methods:

The survey development section is confusing, a clear description of all survey elements in the table is necessary and there are several misleading explanations (for example "5-point Likert scales" and then "Likert scale with three items"). Some population data are show in the table, relevant data are missing and age/sex are mentioned several times (methods, table, and results).

We have made the description clearer, adding all relevant missing information

Results:

The data presentation, especially the abbreviations, is confusing and incomplete. According to previous published studies, for example reference 13 (Serrao C et al.), a clear survey framework with subsequent consistent presentation of all survey results is mandatory.

We have rewritten this section, as suggested, to make it more complete and easy to follow. 

Discussion:

The authors should strictly note the following argument chain in their discussion (in the context of early reports, large studies and complex data analysis): what is already known and published, their own findings and how their findings contribute to existing knowledge.

We revised the discussion section, as suggested.

Reviewer 2 Report

In general, this is a good paper.  However, it can be improved.  First, there are a few mistakes in the text which I have highlighted in the commented version of the paper attached herewith.  Second, the second paragraph of the discussion would fit better in the introduction.  Third, one would expect stronger suggestions for prevention in the conclusion.  Otherwise, this paper can be published after those minor revisions.

Author Response

In general, this is a good paper.  However, it can be improved.

Thank you for your opinion, we have included the requested suggestions to improve the paper.

First, there are a few mistakes in the text which I have highlighted in the commented version of the paper attached herewith. 

We have changed what was requested:

  • insert “so as”, in the introduction: done
  • were all nurses of the institute sent questionnaire? We have made the sentence clearer. The nursing staff in the institute consisted of 48 people. Only 40 answered, but of these 5 did not fill in T1. So the sample consists of 35 nurses.
  • should it be: “All participants”: done.
    • remove “and” in the table 1: done.
  • Inadeguate alignment table 1: done. 
  • Only son or children: We have removed this data, as suggested by another reviewer. 
  • The legend in the table 3 doesn’t match exactly the title in the table: there is a graphic error, we have corrected thethe mistake. 

Second, the second paragraph of the discussion would fit better in the introduction.

We have moved the paragraph, as requested.

Third, one would expect stronger suggestions for prevention in the conclusion.  Otherwise, this paper can be published after those minor revisions.

We have added some considerations in the discussion and conclusion.

Reviewer 3 Report

The article sets out to “assess the emotional state of nurses” working in a judicial psychiatric hospital during the COVID-19 pandemic by conducting online surveys among a confined group of nurses at two points of the pandemics (spring 2020 and late 2021).

Gathering knowledge about nurses working conditions and emotional state is valuable however this article demonstrates several serious shortcomings. Whether these relate to the study design or more simply to the way the data and findings are presented in this article is not possible to assess due to lack of information in the text.

Firstly, despite genuine interest in the topic, this article is not accessible for readers without thorough knowledge about the language, or more precise the abbreviations, of quantitative methods. Even the socio-demographic variables, which are spelled out, leaves me puzzled: I assume (which is not satisfying when reading a paper) that the category “Sons” includes daughters as well, and that this is a translation issue. Still, does the numbers refer to any children the respondents have no matter their age, or does it refer to children with daily care needs only? This is perhaps less important as neither this nor any other sociodemographic issues receive any further attention in the discussion.

What is graver though is the total black-boxing of the questions presented in the online survey to the nurses – on which the results and discussion are based – and of how these questions came about.

Several clinical assessment tools are mentioned and some of them are also very briefly described (section 2.3. and table 2). One assessment tool, DASS-21, does, according to table 2, “measure depression, anxiety, and stress”; another (MBI) “address three different fields of professionalism: [namely] emotional exhaustion, depersonalization, and reduced personal achievement”; a Stress for Covid-19 Questionnaire is also mentioned but not introduced any further. The article sets out, according to the aim stated in the abstract, to assess nurses’ emotional state, but how can these assessment tools, apparently loaded with negative presumptions, uncover any sign of positive emotions?

As the reader have no further insight into the questions, the way they and the project in general were presented to the participants (except that google forms were used), nor to the response alternatives (except that they reportedly were “binary”) we just have to trust whatever the authors claim the codes, or the abbreviations and numbers, presented in section 3, Results, mean.

Based on what I get out of this article, there is a discrepancy between the authors’ aim and claims and the actual process and outcome; the discussion part starts by claiming that the study has “evaluated the mental health of nurses…” but apparently it has evaluated the degree of mental unhealth based on a collection of information on health problems (exhaustion, depersonalization, depression, anxiety and so on).

The survey allegedly included a system usability scale that confirmed that the respondents rendered the usability of the online system – not the questions and response alternatives! – as “high”.

The authors draw three conclusions on separate areas, which are all problematic but not problematized in this article:

Firstly, the authors found “increased stress, anxiety, and burn out staff.” So, they observed an intensification or strengthening of these negative mental conditions but apparently, and as mentioned above, they did apparently not look for any signs of positive mental state or presence or development of positive conditions. Neither does the study say anything about the difference between the study participants and e. g. other work groups or the general population locally or nationally. The fact that the study was conduced among nurses does not automatically prove that the alleged worsening of emotional state at these two points of the covid-19 pandemics was a result of covid-19 related occupational conditions.  

The article argues that the nurses’ mental health “worsened over time” (p.4) but lack of contextual or qualitative information makes it impossible to know whether this is exceptional to nurses, if it is because of covid-related circumstances at the workplace (or other such as reorganization, non-covid related change of the work force or patients/detainees) or whether this reflected a general trend among people in the area.  

Secondly, the authors claim that their study demonstrates that online questionnaires (which contrary to their claim can hardly be characterized as “innovative” anymore) “could be a valid solution to monitor the mental state and working quality of life”. As pointed out above, this study illustrates how the quality of results depends just as much on how tools are used as on the tool itself. How did the online questionnaire, the assessment tools, the analytical tools, and not least the authors contributed to working out the data and conclusions? Furthermore, the idea of monitoring workers mental health evokes several ethical issues that exceed both the article and this review!

Independent from this study, there is no doubt that many people – including nurses –faced increased mental hardship during the covid-19 pandemics. However, and now we face the third conclusion: The bottom line of this study literally prescribes “psychological support to healthcare professionals” as “fundamental to promoting psychological well-being and ensuring adequate service in the health care system, including judicial psychiatry hospitals” (italics added). Here the authors take a fast and dirty shortcut, from symptoms to cure – neglecting all the possible reasons wisely mentioned in the discussion part of the article: The aspects brought in with reference to other studies, such as overcrowded wards, long and inconvenient working hours, staff shortages and psychosocial problems among the detainees, are all likely and relevant, but to our knowledge these are unexplored reasons for the worsening emotional state among these respondents.

The prescribed cure, “psychological support to healthcare professionals”, might contribute to “promoting psychological well-being” among the nurses, but there are no findings in the material presented in this article that support the concluding argument that this (alone) is “fundamental”. Based on a substantial amount of qualitative and quantitative research on care work, some mentioned in the discussion part of this article, it is mainly structural and organizational conditions that cause mental, emotional, and physical hardship among nurses and care workers in general (among the respondents the relation seems as mentioned unexplored) and it has been documented that covid-related issues have intensified these structural shortcomings and their effects.

Prescribing psychological support to the staff this article contributes to displace the responsibility from the structural to the individual level, onto the individuals; the article places the problem and site of reparation (by the means of psychological support) at the staff – thereby naturalize and legitimize the systems that causes mental and emotional stress. Hence, most likely unintendedly and most certainly unfoundedly, this article sustains the political and institutional machinery that produces burnout.

Author Response

The article sets out to “assess the emotional state of nurses” working in a judicial psychiatric hospital during the COVID-19 pandemic by conducting online surveys among a confined group of nurses at two points of the pandemics (spring 2020 and late 2021).

Gathering knowledge about nurses working conditions and emotional state is valuable however this article demonstrates several serious shortcomings. Whether these relate to the study design or more simply to the way the data and findings are presented in this article is not possible to assess due to lack of information in the text.

Thank you for your evaluation on our paper, we made the required changes adding all missing information.

Firstly, despite genuine interest in the topic, this article is not accessible for readers without thorough knowledge about the language, or more precise the abbreviations, of quantitative methods. Even the socio-demographic variables, which are spelled out, leaves me puzzled: I assume (which is not satisfying when reading a paper) that the category “Sons” includes daughters as well, and that this is a translation issue. Still, does the numbers refer to any children the respondents have no matter their age, or does it refer to children with daily care needs only? This is perhaps less important as neither this nor any other socio-demographic issues receive any further attention in the discussion. 

We have rebuilt methods by reducing/specifying abbreviations and adding the missing information. The child-related aspect was removed as suggested also by another reviewer. 

What is graver though is the total black-boxing of the questions presented in the online survey to the nurses – on which the results and discussion are based – and of how these questions came about. 

Several clinical assessment tools are mentioned and some of them are also very briefly described (section 2.3. and table 2). One assessment tool, DASS-21, does, according to table 2, “measure depression, anxiety, and stress”; another (MBI) “address three different fields of professionalism: [namely] emotional exhaustion, depersonalization, and reduced personal achievement”; a Stress for Covid-19 Questionnaire is also mentioned but not introduced any further. The article sets out, according to the aim stated in the abstract, to assess nurses’ emotional state, but how can these assessment tools, apparently loaded with negative presumptions, uncover any sign of positive emotions? 

The tests are standardized and commonly used in clinical practice to evaluate the emotional and psychological states of the subjects: Nonetheless,  after having included the references of the tests used, we have also expanded the explanation of the tests. 

As the reader have no further insight into the questions, the way they and the project in general were presented to the participants (except that google forms were used), nor to the response alternatives (except that they reportedly were “binary”) we just have to trust whatever the authors claim the codes, or the abbreviations and numbers, presented in section 3, Results, mean. 

Based on what I get out of this article, there is a discrepancy between the authors’ aim and claims and the actual process and outcome; the discussion part starts by claiming that the study has “evaluated the mental health of nurses…” but apparently it has evaluated the degree of mental unhealth based on a collection of information on health problems (exhaustion, depersonalization, depression, anxiety and so on). 

We have specified the tests better and inserted the alternative answers and examples of the questions administered to the sample. Furthermore, we have included the references of the articles in which the Italian validation of the instruments was carried out, with the full items of the questionnaire and the aims of the questionnaire. Finally, we have better specified the burnout indices supported by Mashlach and evaluated by the MBI questionnaire already in the introduction and inserted an adequate explanation of the other tests.

The survey allegedly included a system usability scale that confirmed that the respondents rendered the usability of the online system – not the questions and response alternatives! – as “high”. 

We have better specified what sus was used for, and what kind of usability we were going to perform, with a more detailed explanation in table 2. It is our opinion that if the user does not know the online tool, perceives it as unusable, and has difficulty entering the answer, the data obtained are falsified. The presence of the SUS allows evaluating if there is a bias connected to whether the questionnaire was not filled in correctly due to technical problems and poor ability to use the online forms.

The authors draw three conclusions on separate areas, which are all problematic but not problematized in this article:

Firstly, the authors found “increased stress, anxiety, and burnout staff.” So, they observed an intensification or strengthening of these negative mental conditions but apparently, and as mentioned above, they did apparently not look for any signs of a positive mental state or presence or development of positive conditions. Neither does the study say anything about the difference between the study participants and e. g. other work groups or the general population locally or nationally. The fact that the study was conducted among nurses does not automatically prove that the alleged worsening of emotional state at these two points of the covid-19 pandemics was a result of covid-19 related occupational conditions.  

Thanks for the comment. Our study focused on nurses in prisons. We put within the boundaries of the study that there was no comparison with other populations because it was beyond our goals. However, we updated the literature to compare the data obtained in different populations. It is noteworthy that we have considered some positive indices, such as personal fulfillment. Surely, further studies, also considering the coping strategy or other factors, are needed in the future. 

The article argues that the nurses’ mental health “worsened over time” (p.4) but lack of contextual or qualitative information makes it impossible to know whether this is exceptional to nurses, if it is because of covid-related circumstances at the workplace (or other such as reorganization, non-covid related change of the work force or patients/detainees) or whether this reflected a general trend among people in the area.  

Thanks for the suggestion, we added this aspect in the limitation of the study section. 

Secondly, the authors claim that their study demonstrates that online questionnaires (which contrary to their claim can hardly be characterized as “innovative” anymore) “could be a valid solution to monitor the mental state and working quality of life”. As pointed out above, this study illustrates how the quality of results depends just as much on how tools are used as on the tool itself. How did the online questionnaire, the assessment tools, the analytical tools, and not least the authors contributed to working out the data and conclusions? Furthermore, the idea of monitoring workers mental health evokes several ethical issues that exceed both the article and this review!

The questionnaires, being validated and standardized, have a coding system that allows for reducing the bias of "subjective interpretation" of the examiners. The online questionnaires were composed of standardized self-administration questionnaires to reduce this bias. In addition, the presence of the SUS to assess the perception of the subject on the use of the online form allows you to have greater certainty that the answers given by the subjects represent their mental dispositions.

The participants all gave their consent for the research and publication of the results, so we tried to respect their will ethically.

Obviously there are various limits, as in any exploratory research, even more so that we are interfacing on such a complicated topic.

Independent from this study, there is no doubt that many people – including nurses –faced increased mental hardship during the covid-19 pandemics. However, and now we face the third conclusion: The bottom line of this study literally prescribes “psychological support to healthcare professionals” as “fundamental to promoting psychological well-being and ensuring adequate service in the health care system, including judicial psychiatry hospitals” (italics added). Here the authors take a fast and dirty shortcut, from symptoms to cure – neglecting all the possible reasons wisely mentioned in the discussion part of the article: The aspects brought in with reference to other studies, such as overcrowded wards, long and inconvenient working hours, staff shortages and psychosocial problems among the detainees, are all likely and relevant, but to our knowledge these are unexplored reasons for the worsening emotional state among these respondents. 

Thank you for your opinion, surely the goal of our study is complex and we have evaluated only part of the problem. We do not think that psychological support alone solves the difficulties, compared to more complex measures that improve the situation drastically. We improved the discussion by including your suggestions.

The prescribed cure, “psychological support to healthcare professionals”, might contribute to “promoting psychological well-being” among the nurses, but there are no findings in the material presented in this article that support the concluding argument that this (alone) is “fundamental”. Based on a substantial amount of qualitative and quantitative research on care work, some mentioned in the discussion part of this article, it is mainly structural and organizational conditions that cause mental, emotional, and physical hardship among nurses and care workers in general (among the respondents the relation seems as mentioned unexplored) and it has been documented that covid-related issues have intensified these structural shortcomings and their effects. 

Prescribing psychological support to the staff this article contributes to displace the responsibility from the structural to the individual level, onto the individuals; the article places the problem and site of reparation (by the means of psychological support) at the staff – thereby naturalize and legitimize the systems that causes mental and emotional stress. Hence, most likely unintendedly and most certainly unfoundedly, this article sustains the political and institutional machinery that produces burnout.

We proceeded to modify the sentence to avoid misunderstanding, and we have now specified that improving structural and organizational conditions may reduce work stress and, in specific cases, psychological support could be of help.

Round 2

Reviewer 3 Report

It was a pleasure to read this revised version! Well done!